

# Initiation of a major calving event on Bowdoin Glacier captured by UAV photogrammetry

Guillaume Jouvet[1], Yvo Weidmann[1], Julien Seguinot[1], Martin Funk[1], Takahiro Abe[2], Daiki Sakakibara[3], Hakime Seddik[3], and Shin Sugiyama[3]

[1]ETHZ, VAW, Zurich, Switzerland
[2]Graduate School of Science, Hokkaido University, Sapporo, Japan
[3]Institute of Low Temperature Science, Hokkaido University, Sapporo, Japan

*Correspondence to:* jouvet@vaw.baug.ethz.ch

**Abstract.**

In this paper, we analyse the calving activity of Bowdoin Glacier, north-west Greenland, in 2015 by combining satellite images, UAV (Unmanned Aerial Vehicles) photogrammetry and ice flow modelling. In particular, a high-resolution displacement field is inferred from UAV orthoimages taken immediately before and after the initiation of a large fracture, which induced a

major calving event. A detailed analysis of the strain rate field allows us to map accurately the path taken by the opening crack. Modelling results reveal i) that the crack was more than half-thickness deep, filled with water, and getting irreversibly deeper when it was captured by the UAV and ii) that the crack initiated in an area of high horizontal shear caused by a local basal bump immediately behind the current calving front. The asymmetry of the bed at the front explains the systematic calving pattern observed in May and July-August 2015. As a corollary, we infer that the calving front of Bowdoin Glacier is currently stabi-

lized by this bedrock bump and might enter in an unstable mode and retreat rapidly if the glacier keeps thinning in the coming years. Beyond this outcome, our study demonstrates that the combination of UAV photogrammetry and ice flow modelling is a promising tool to track horizontally and vertically the propagation of fractures responsible for large calving events.

## 1 Introduction

In the last decades many ocean-terminating outlet glaciers of the Greenland Ice Sheet (GIS) experienced thinning and rapid

retreat (Joughin et al., 2010; Pritchard et al., 2009), which contributed to the global loss of ice and affected sea level rise. Approximately half of the ice ablation of the GIS is due to calving, i.e. the release of icebergs at the edge of glaciers (Enderlin et al., 2014). The calving mechanism is still not entirely understood mostly because of the complex interconnection between involved processes (Benn et al., 2007b). Among them, the sharp acceleration of the ice flow at the calving front in response to high buoyant forces generates deep crevasses, which precondition the future breaking-off of icebergs. Since calving events are

triggered by fracturing processes, calving is often considered as a random process, which must be analysed over large data sets of events, and relatively few studies have been dedicated to a description of calving at the level of individual events (e.g. O'Neel et al., 2003; Chapuis and Tetzlaff, 2014, and references therein). However, observations in Greenland, (i.e., Medrzycka et al.,



2016), show that large-scale, occasional calving events often contribute more to total calving loss than small-scale frequent events.

Calving primarily results from the propagation of fractures upstream the calving front in response to high stresses (Van der Veen, 1998). Opening and sustained growth of cracks occur when the normal and the shear stress components exceed a certain threshold (Colgan et al., 2016) although the normal component is often assumed to be a primary control over the shear one (Benn et al., 2007b). For a given normal strain rate, Nye (1957) derived a function for calculating the depth of a crevasse by assuming that it will extend until the normal strain rates balance the creep closure rate resulting from the ice overburden pressure. Based on this function, Benn et al. (2007a) derived a calving law by assuming that calving occurs when a crevasse penetrates the water line. Earlier, Vieli et al. (2001) proposed another criterion based on the height-above-buoyancy. However, these semi-empirical approaches assume closely-spaced crevasses and do not account for the stress concentration at the tip of cracks. To overcome this problem, one has to implement more complex continuous approaches based on linear elastic fracture mechanics (Van der Veen, 1998), or continuum damage mechanic (Pralong and Funk, 2005), which merges in the same framework the micro-scale formation of cracks and the macro-scale behaviour of the damaged ice flow. Despite a large number of contributions on calving modelling in recent years, relatively few studies have attempted to model calving events individually (Aström et al., 2013), presumably because microscopic approaches can hardly be coupled to traditional ice flow unlike macroscopic approaches.

The main obstacle to implement continuous models is the lack of data to constrain parameters related to the fracturing of ice (Pralong and Funk, 2005). In particular, no direct measurements of failure stress values within the ice are available except for laboratory ones which, however, cannot reproduce the observed orders of magnitude. By contrast, critical strain rates can be measured. Although $0.01 \, a^{-1}$ is often given as reference threshold (Meier, 1958), observed threshold normal strain rates for the initiation of crevasses span over two order of magnitude (Colgan et al., 2016). The estimation of critical strain rates has been rendered easier through the use of automatic camera and aerial photogrammetry combined with feature tracking techniques (Colgan et al., 2016; Messerli and Grinsted, 2015). In particular, recent developments of "Unmanned Aerial Vehicles" (UAV) and photogrammetry by structure-from-motion allows high-resolution ice flow velocity fields and associated strain rates to be inferred (Ryan et al., 2015).

In this paper, we use UAV photogrammetry, feature tracking and ice flow modelling to analyse in detail the propagation of a major crack on Bowdoin Glacier, north-west Greenland, before it collapses and generates a major calving event representing about 5% of the annual calving in term of area. From this analysis, we investigate the calving pattern at the terminus of Bowdoin Glacier in May and July-August 2015 observed on satellite images. This paper is organized as follows. First, we describe our data and the methods we have employed to lead this study. Then we present our results related to satellite images, UAV-inferred ice flow field, strain analysis, and ice flow modelling. Finally, we use these results to revisit and explain the calving pattern of Bowdoin Glacier observed in 2015.



## 2 Data and methods

### 2.1 Study site

Bowdoin Glacier (77°41' N, 68°35' W) is an ocean-terminating glacier, which belongs to a network of outlet glaciers located in the north-west of the Greenland ice sheet. It is approximately 10 km long and 3 km wide with an average surface slope of less than 1°. The glacier discharges into the Bowdoin fjord through a 3 km wide most likely grounded calving front (Sugiyama et al., 2015). At the center of the calving front, Bowdoin Glacier was about 250 m thick, and its maximal flow speed was about 1.5 m d$^{-1}$ in 2013. A medial moraine, that we often use as a reference line in what follows, runs parallel to and about 1 km away from the left glacier margin (see Fig. 1).

The front of Bowdoin Glacier has been fairly stable since the end of the 19th century. Its present-day position is no more than 2 km upstream its 1897 position (Chamberlin, 1897). However, from July 2008 to September 2013, a rapid retreat of the calving front at a rate of 220 m a$^{-1}$ was recorded, while no significant changes occurred during the previous 20 years (Sugiyama et al., 2015). From 2013, the calving front stabilized to about its current position. The rapid retreat recorded during the period 2008-2013 was attributed to a local depression in the bedrock, which is known to make the grounding line unstable (Schoof, 2007).

In July 2013, 2014 and 2015, we carried out field measurements on Bowdoin Glacier. This included ice radar and GPS measurements (Sugiyama et al., 2015; Tsutaki et al., 2016), seismic records (Podolskiy et al., 2016), automatic camera installations, and boreholes drilling (to record internal ice deformation, englacial temperature, and water pressure). As a relevant outcome for this study, we found that the ice deformation represents less than 10% of the total motion in two boreholes located about 2 km upstream the calving front (Seguinot et al., in prep.).

### 2.2 Satellite images

To analyse the large-scale calving pattern and ice flow field of Bowdoin Glacier in 2015, we used the Landsat 8 Operational Land Imager panchromatic images. In particular, these data allowed us to infer the positions of the calving front (see Fig. 1) and estimate the calved surface area during the largest events.

### 2.3 UAV photogrammetry

During the 2015 field campaign, we flew an "Unmanned Aerial Vehicle" (UAV) over the terminus of Bowdoin Glacier to collect aerial images and reconstruct in high resolution the 3D geometry of the calving front by photogrammetry. Although we closely followed the methods described by Ryan et al. (2015), we shortly review the techniques we have employed from UAV implementation to image post-processing.

As UAV, we used the 2-meter-wide fixed-wing "Skywalker X8" equipped with the "Pixhawk" open-source autopilot (https://pixhawk.org/) running with the APM "artduplane" firmware (http://ardupilot.org/ardupilot/). The UAV also carried a camera (Sony Alpha 6000), which had a 24 megapixel sensor, and a 16mm lens. The UAV flew twice autonomously (on the 11th and





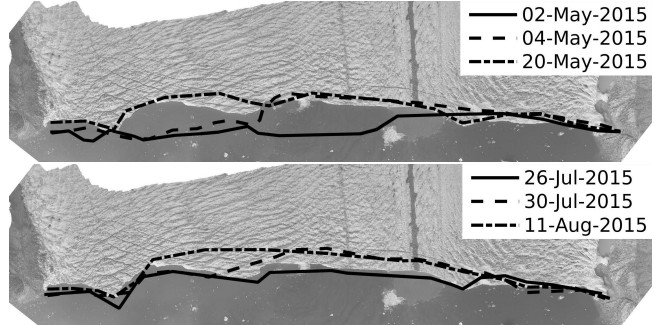

**Figure 1.** Positions of the calving front of Bowdoin Glacier inferred from satellite images before and after the calving events of May 3, May 19, July 27 and August 9.

the 16th of July) following a pre-programmed sequence of waypoints located at the two extremities of the glacier front so that the calving front was covered by four parallel flight lines. For each flight, the UAV flew 25 km at an altitude of about 300 meters above the ground, and collected about 1000 overlapping pictures of the calving front with a resolution of 7 cm per pixel. An overlap of 95% in flight direction and 70% in cross flight direction was chosen to obtain a stable aerotriangulation.

For a reliable georeferencing Ground Control Points (GCP) were installed on the left border of the glacier and on the moving glacier surface. The position of the moving GCPs was recorded repeatedly with static DGPS positioning system so that their absolute positions at the time of each flight was linearly approximated. The pictures of each flight were post-processed through the software Agisoft PhotoScan (http://www.agisoft.com/). Using the image correlation techniques (structure-from-motion) of Photoscan, a high-resolution 3D model and orthoimage (covering about 2 km$^2$) of the calving front was reconstructed for each

flight (see Figs. 2a and 2b).

## 2.4  Feature tracking

The orthoimages derived from satellite and UAV data are used to infer the surface displacement fields of the ice by feature tracking method. There exist a variety of image matching methods to derive glacier surface speed (Heid and Kääb, 2012). Here, we have used an algorithm of cross-correlation in frequency domain on orientation images algorithm (Abe et al., 2016; Heid

and Kääb, 2012; Sugiyama et al., 2015). On the one hand, a 30-meter resolution velocity field averaged between June 14 and August 1 (see Fig. 3) was inferred from satellite images. On the other hand, from a native resolution of 10 cm for the UAV orthophoto, a template searching chip of 200x200 pixels and a step number of 10x10 pixels, we obtained a 1-meter resolution velocity field averaged between July 11 and July 16 (see Fig. 2c).





**Figure 2.** Calving front of Bowdoin Glacier: Orthoimages obtained by UAV on the 11th (a) and the 16th (b) of July, resulting velocity field inferred by a feature tracking (c), and maximum principal directions of the strain rate (d), respectively. For the sake of visualisation, only the maximum directions with a magnitude above $5\ a^{-1}$ are drawn on d). The highest magnitudes are reached along the main crack, and correspond to maximal principal strain rate of about $150\ a^{-1}$.



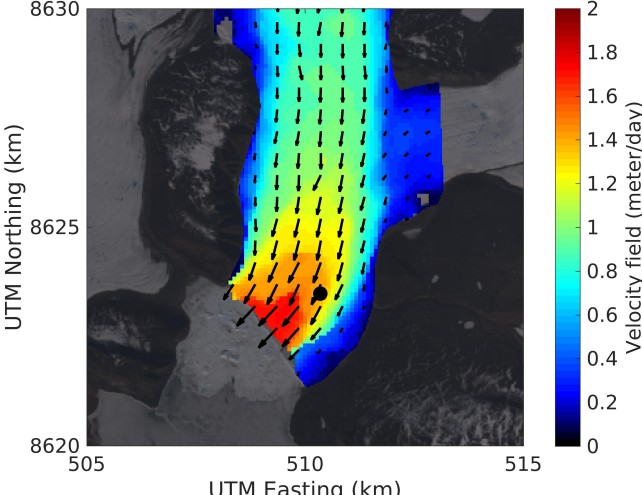

**Figure 3.** Average velocity field between June 14 and August 1 inferred by feature tracking from LANDSAT satellite images. The back dot indicates the position of the borehole.

## 2.5   Strain analysis

The first-order control for triggering calving events is related to the strain rates (Benn et al., 2007b). For this reason, we computed the horizontal part of the strain rate tensor:

$$
D = \begin{pmatrix} \partial_x u_x & (\partial_y u_x + \partial_x u_y)/2 \\ (\partial_y u_x + \partial_x u_y)/2 & \partial_y u_y \end{pmatrix},
$$

where the derivatives are approximated by finite difference, from the horizontal velocity field $(u_x, u_y)$ inferred by feature tracking. However, $D$ depends on the system of coordinate $(x, y)$, and thus cannot be interpreted directly. Instead, we computed the eigenvalues of $D$ since they do no depend on the system of coordinate. The highest eigenvalue (called the maximal principal strain) indicates the maximal normal strain rate among all possible directions, while its associated eigenvector (called the maximal principal direction) is the direction in which the normal strain is maximized (see Fig. 2d).

## 2.6   Modelling

The UAV can only capture the ice dynamics at the glacier surface. To gain insight into the dynamical processes within the glacier, the terminus of Bowdoin Glacier was modelled using the Elmer/Ice code (Gagliardini et al., 2013). To do so, we built a 3D mesh of the glacier front from estimated basal and surface ice topographies using the GMSH mesher (Geuzaine and Remacle, 2009). The basal topography was extrapolated with a parabolic profile from ice radar measurements (Sugiyama

et al., 2015) while the surface topography was derived from the UAV-inferred 3D model, and completed by a larger available Digital Elevation Model. Elmer/Ice implements a full-Stokes model (Greve and Blatter, 2009), and ice is considered as a





Non-Newtonian fluid governed by Glen's flow law:

$$\epsilon_{ij} = EA(T){\sigma_{II}}^{n-1}\sigma_{ij}, \tag{1}$$

where $\epsilon$ and $\sigma$ denote the strain and deviatoric stress tensor, $\sigma_{II}$ denotes the second invariant of $\sigma$, $A$ is the temperature-dependent Arrhenius factor (Cuffey and Paterson, 2010), and $E$ is an enhancement factor, which controls the stiffness of ice.

A relationship between englacial temperature and depth was obtained from borehole measurements (Seguinot et al., in prep.), and generalized over the whole modelled domain. The model was supplemented by the following boundary conditions. No force applies on the top ice surface,

$$\sum_i \sigma_{ij} n_i = 0, \tag{2}$$

while the following condition applies at the calving front:

$$\sum_i \sigma_{ij} n_i = \rho_w g \min(z,0)\, n_j, \tag{3}$$

where $(n_x, n_y, n_z)$ is the outer normal vector. At the back and lateral boundaries of the modelled domain, we impose the Dirichlet boundary condition,

$$u_i = u_{i,\mathrm{meas}}, \tag{4}$$

where $u_{i,\mathrm{meas}}$ equals the UAV-inferred velocity field at the glacier surface, and decreases linearly with the depth to reach $90\%$

of the surface motion at the bedrock consistently to borehole measurements of the ice deformation. At the glacier bed, one applies Budd's friction law (Budd et al., 1984), which links basal shear stress $\tau_b$ to basal sliding $u_b$ through the relation:

$$\tau_b = C u_b^{1/3}(1-f)^\alpha, \tag{5}$$

where $C > 0$ is a sliding coefficient, $\alpha > 0$ is a given exponent, and $f = (\rho_w/\rho_i) \times (-b/h)$ is the floatation ratio (dimensionless), which is $0$ when no buoyant force applies and $1$ when ice is floating (this is never the case in the present study), $\rho_w$ and

$\rho_i$ denote the density of water and ice, $b$ the bedrock elevation, and $h$ the ice thickness. Since only the terminus of the glacier was modelled, the buoyancy under the glacier bed could be reasonably calculated as a simple function of depth $z$ assuming a fully distributed hydrological system. In that case, the effective pressure $N$, which is defined by the ice overburden pressure minus the buoyancy, is related to the floatation ratio $f$ through the relation $N = \rho_i g h (1-f)$. Thus, Eq. (5) can be understood as follows: a high buoyancy renders the effective pressure $N$ or $(1-f)$ small and favours sliding. Parameter $\alpha$ in Eq. (5)

therefore controls the degree of influence of the floatation ratio on sliding from no-influence when $\alpha = 0$ (Weertman's law) to a quadratic influence when $\alpha = 2$.

The model was run in two-steps. First, ice flow parameters $E$, $\alpha$, and $C$ in Eq. (1) and (5) were tuned so that i) the modelled surface velocities match as good as possible the measured ones ii) the ratio between vertical ice deformation and total ice motion at the borehole is about $10\%$, as measured in 2014-2016 by borehole inclinometers (Seguinot et al., in prep.) (see Fig.





4, top panel). Since the borehole is located about 2 km upstream the calving front (see Fig. 3), we first modelled an enlarged domain, and used satellite-inferred velocity field (see Fig. 3) instead of the UAV-inferred ones since it covers a larger area. Once the three ice flow parameters were tuned, we restricted our domain to the glacier front, and optimized (to improve the consistency between modelled and measured velocity fields) according to two geometrical parameters $h_{\text{lift}}$, and $d_{\text{frac}}$ of the

glacier front (see Fig. 4, middle and bottom panels). First, the bedrock topography was lifted laterally from the the moraine position to the south-east glacier side to mimic with a presumable bump responsible for the sharp transition between fast and slow ice flow. The height of this bump is denoted $h_{\text{lift}}$. Second, a cut of constant depth $d_{\text{frac}}$ was applied over the surface at the location of the main fracture to simulate the real depth, which cannot be inferred from aerial images. It must be stressed that this cut directly affects the mesh. Thus, opposed hydrostatic forces (boundary condition (3)) apply to both sides of the fracture

(assuming it is filled of water up to sea level, see Section 4) with the result of intensifying its opening.

## 3   Results

### 3.1   Calving pattern

The analysis of the 2015 satellite images reveals that substantial ablation by calving was due to few monthly-spaced calving events. In particular, four May and July-August events (which occurred on May 3, May 19, July 27 and August 9) contributed

by 0.13, 0.13, 0.17 and 0.07 km$^2$ of surface loss, respectively (see Fig. 1). These numbers must be compared to the annual loss by calving that we estimate to $1.2 \pm 0.2$ km$^2$. This estimate was obtained by integrating the ice flux across the calving front from the satellite-inferred velocity field (see Fig. 3), the calving front being fairly stable since 2013. In total, the four events contributed to about 20% of the annual calving. Intriguingly, the events of May and July/August show similar patterns: a first slice of ice splits on the south-east side, however, without touching the glacier border, and, a second slice separates further

north-west a tens of days later (see Fig. 1).

### 3.2   Calving event of the 27th July

During the summer 2015 field campaign, field observations, and UAV images allowed us to reconstruct step-by-step the early stages preceding the calving event, which occurred on July 27. On July 11, the UAV orthoimage shows no major fracture in the vicinity of the calving front, and gives no signs that a major event is about to trigger (see Fig. 2a). One day later, team members

of the field campaign reported that a few-meter-wide crack appeared about 100 meters upstream the calving front across the medial moraine. On July 13, the crack was visible from an automatic camera, which monitors the calving front at about 2 km distance. On July 16, the UAV orthoimage clearly indicates that the crack was about 750 meters long (see Fig. 2b). On July 27, an about 1-km-long slice of the front (i.e. about one third) collapsed (see Fig. 1).





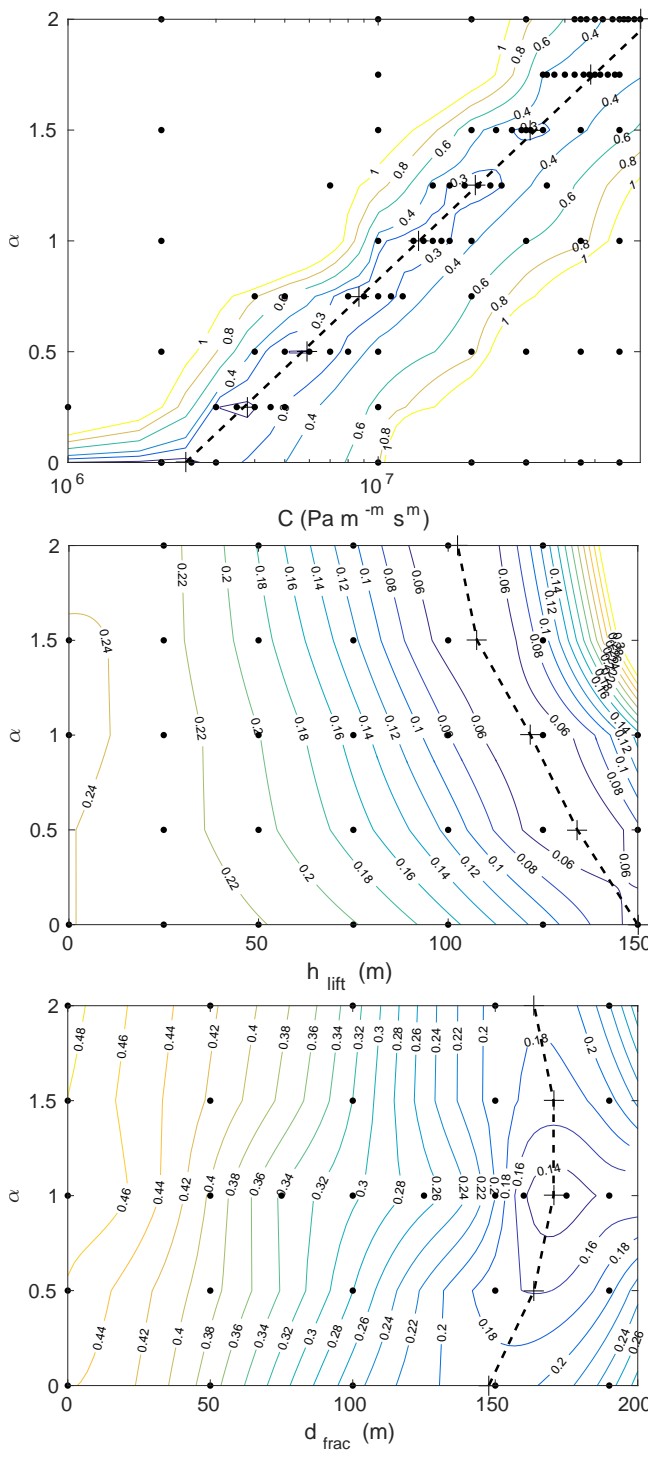

**Figure 4.** Level sets of the absolute misfit (in meters per day) between modelled and measured surface velocity magnitudes as function of parameters $\alpha$ (Y axis) and $C$, $h_{\text{lift}}$, and $d_{\text{frac}}$ (X axis), respectively. Each black dot corresponds to one model realization, from which the level sets were extrapolated. For each value of $\alpha$, the symbol + displays the minimum with respect to $C$, $h_{\text{lift}}$, or $d_{\text{frac}}$, respectively, while the dashed line connects + symbols.





### 3.3 Velocity and strain rate fields

Figures 3 and 2c show the horizontal ice flow velocity fields inferred by feature tracking from satellite and UAV orthoimages, respectively. Although the satellite-inferred data covers a larger domain and time period than the UAV-inferred one, we obtain a good agreement where the two results overlap. Interestingly, the UAV result shows a clear discontinuity along the crack, which was identified on the orthoimage of July 16 (see Fig. 2b). Additionally, Figs. 3 and 2c reveal a sharp transition in the ice flow speed: homogeneously fast flow in the central section and very slow motion in the south-eastern part within about 700 m from the glacier's margin. Finally, Fig. 2d displays the maximum strain rate principal directions computed from the velocity field. As a matter of fact, the main fracture and the highly damaged zone delimiting the slow and fast bands concentrate the highest maximal principal strain rates.

### 3.4 Modelling results

First, we calibrated the ice flow and sliding parameters $E$, $C$ and $\alpha$ (related to Eq. (1) and (5)) to simultaneously minimize the misfit between modelled and measured surface velocities and to match the deformation rate of ice recorded at the borehole. In fact, we found that the enhancement factor $E$ could be tuned independently of the other parameters and the value $E = 10$ proved to give a consistent deformation rate at the borehole (i.e. about $10\%$ of vertical ice deformation) irrespectively to other parameters. As a matter of fact, $E = 10$ is a rather large enhancement factor. Yet, this value has little influence on the results since the ice deformation is little compared to basal sliding. For each $\alpha \in \{0, 0.25, 0.5, 0.75, 1, 1.25, 1.5, 1.75, 2\}$, we found a unique parameter $C$ minimizing the misfit between modelled and measured surface velocities (see Fig. 4, top panel). However, it was not possible to clearly identify the best couple $(C, \alpha)$ since all combinations give comparable misfits (about 0.2-0.3 meters per days). Finally, we found a good linear fit between $\alpha$ and the logarithmic of the sliding coefficient $C$ through the following empirical relationship:

$$C \approx 2.4 \times 10^6 \times e^{1.73\alpha} \, \mathrm{Pa\,m^{-1/3}\,s^{1/3}}, \tag{6}$$

which is in the same order of magnitude than the value $C = 6.06 \times 10^4 \times h^\alpha \, \mathrm{Pa\,m^{-1/3}\,s^{1/3}}$ obtained for Columbia Glacier in Nick (2006) for $\alpha = 3.5/3$ when the ice thickness $h$ spans between 100 and 300 meters.

Second, we optimized the bump height $h_{\mathrm{lift}}$ on the south-east side of the glacier front (see Fig. 5) to reproduce at best the high shear zone shown by the UAV-inferred velocity field (see Figs. 2c and 6). Figure 4 (middle panel) displays the misfit between modelled and measured surface velocities as function of $\alpha$ (and the optimal $C$ given by Eq. (6)) and the bump height $h_{\mathrm{lift}}$. Note that the measure of the misfit excludes the detaching slice of ice to better focus on the high shear zone. Interestingly, as in the previous optimisation, one cannot clearly constrain $\alpha$ from this optimisation since several values give comparable misfits (see Fig. 4, middle panel). However, we observe that bumps are slightly thinner with high values of $\alpha$. Indeed, basal motion contributes to most ($90\%$) of the total motion so that a bump on the bed slows down further the ice flow when the influence of the reduced effective pressure $N$ on basal sliding is the highest (i.e. for high $\alpha$). As a result we found that a $h_{\mathrm{lift}} = 125$ meter-high bump was necessary to generate a high shear zone comparable to the one derived by UAV when using



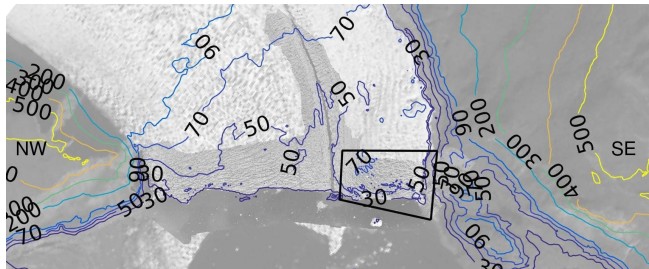

**Figure 5.** Contour lines of the Digital Elevation Model in the neighbourhood of the calving front. The polygon indicates the place where the bedrock was lifted in modelling experiments. The plot reveals the presence of a 20-meter-high bump on the glacier surface inside the polygon, and of a ridge on the south-east side of the glacier.

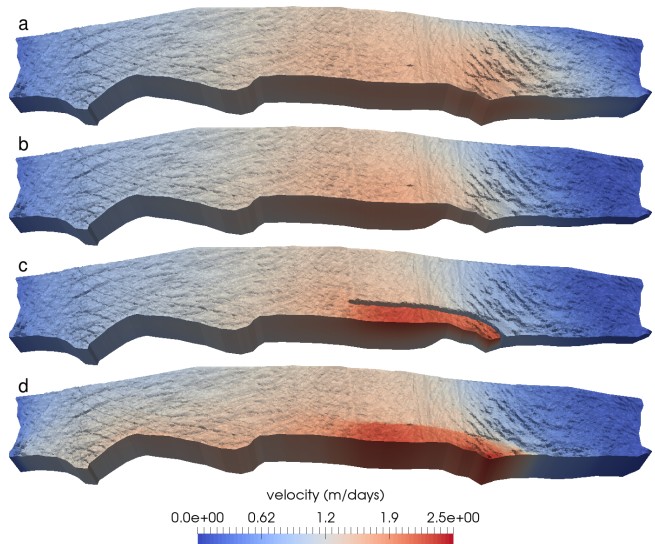

**Figure 6.** Magnitude of the modelled velocity field after the first (a), the second (b) and the third (c) optimisation, i.e. a) corresponds to the original setup with optimal parameters $(E, C, \alpha)$ b) accounts for the optimal bump height on the bed while c) accounts for the optimal crack depth. For the sake of comparison, d) displays the measured UAV-inferred velocity field on the surface.

the mean $\alpha = 1$, with an uncertainty of $\pm 25$ m when accounting the limit cases $\alpha = 0$ and $\alpha = 2$. After including such a bump on the bedrock, the misfit dropped from 0.2-0.3 to 0.05 meters per days (as compared to the first optimisation) for any value of $\alpha$.

Third, we optimized the depth of the main fracture $d_{\mathrm{frac}}$ to reproduce at best the discontinuity shown by the UAV-inferred velocity field at the crack location, assuming a $h_{\mathrm{lift}} = 125$ meter-high bump over the bedrock. Figure 4 (bottom panel) displays the misfit between modelled and measured surface velocities as function of $\alpha$ (and $C$ given by Eq. (6)) and the fracture depth



$d_{\text{frac}}$. The measure of the misfit is restricted to the area near and downstream the crack. As a result, only a $160 \pm 10$-meter-deep crevasse shows a result consistent with UAV-inferred observations (see Figs. 4, bottom panel, and 6). Interestingly, this result is fairly independent of the considered sliding law parametrization ($\alpha$ and $C$). As a matter of fact, including the crevasse depth in the model reduced the misfit by more than 65% (see Fig. 4). Additional modelling setup based on non-constant crack depths
$d_{\text{frac}}$ were run. However, constant depth runs yielded to the best match between modelled and measured velocities (results not shown).

## 4   Discussion

The orthoimage of the 16th of July reveals a main fracture (Fig. 2b), however, it is not clear whether the fracture reaches the calving front or not due to the presence of seracs at the glacier surface. The discontinuity in the velocity field (Fig. 2c)
along the fracture settles this issue, confirms that the fracture extents to the front, and describes accurately its path. This discontinuity suggests that a slice is detaching from the glacier on the south-eastern side, and is only hanging to the other side. Corroboratively, Figure 2d shows very high maximal principal strain rate components near the crack (between 50 to 150 a$^{-1}$). As a matter of fact, the directions of maximal strain along the crack are not always perpendicular to the fracture, but tilt up to 45° toward the centerline of the glacier. This indicates that without control from the south-eastern side, the entire slice
tends to rotate around its extremity, which is still anchored. This result is confirmed by modelling results: only an about 160-meter-deep and water-filled crevasse (i.e. about two thirds of the ice thickness) can reproduce the observed jump in velocities (see Figs. 4, bottom panel, 6, and 7). Additional modelling results showed that the crack depth is rather constant along the crack path. Because the crack was connected to the sea (see Fig. 2a), the water level within the crevasse most likely coincided with the sea level, or was slightly above. In that configuration, we can estimate the pressure exerted by ice and water at the
bottom of the crack to about $1.41$ MPa and $1.27$ MPa, respectively. On the other side, the tensile force must have been in the order of magnitude of critical stress thresholds, i.e. around $0.1$-$0.3$ MPa (Vaughan, 1993). Yet, the crack is expected to deepen further if the pressure of water plus the tensile force exceed the pressure of ice (Van der Veen, 1998). As a result of this delicate balance, we can reasonably assume that the crack propagated downward after the 16th of July in response to small perturbations, like for instance a slight increase of the water level within the crevasse. The deepening of the crack must have
increased further the stress concentration at the crack tip, and caused the crack to extend further laterally before to collapse (see Fig. 1). Unfortunately, we can hardly corroborate our crack depth value with Nye's formula (Nye, 1957), which provides an analytic expression for the crevasse depth as a function of the tensile strain and the depth of water in the crevasse, since the Nye's formula does not account for stress concentration effects, which are significant in the present situation.

The velocity field displayed on Fig. 2c clearly shows that there is a highly sheared zone located upstream the junction
between the crack and the calving front, the velocities experiencing a strong variation between slow/resistive lateral and fast/floating central flow. Such a pattern is typical for glacier margins, where the lateral drag induced by the fjord's buttressing is the highest. However, this zone is not directly in contact with the glacier margin, but is located between 500 to 1000 m away from it. In addition, this area is characterized by many large crevasses, which are 45° tilted with respect to the flow direction,



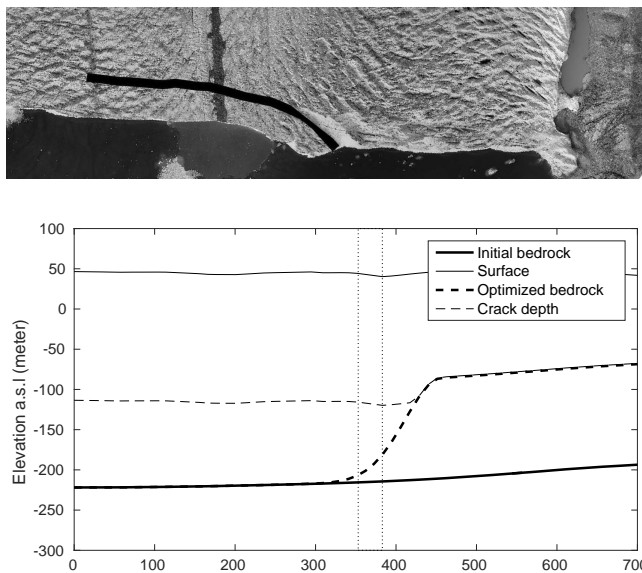

**Figure 7.** Top: Inferred position of the crack on the orthoimage of July 16. Bottom: Bedrock and ice surface along the main crack starting from the tip before and after optimizing the bump height and the crack depth to the UAV-inferred ice flow velocities. The dotted rectangle indicates the position where the crack crosses the moraine.

a typical characteristic of highly sheared margin zones (Paterson, 1999; Colgan et al., 2016). Such a situation can only be the result of a strong irregularity of the bedrock or of the basal conditions. Ice radar measurements of the basal topography across a profile about 800 meters upstream the calving front indicates a rather flat bed in the central part and a constant slope profile between the medial moraine and the margin (Sugiyama et al., 2015). Such configuration might cause transversal gradients of the velocity (the central part of the glacier being nearly floating while the margin experiences more basal resistance), but, not to the extent shown by Fig. 2c. Unfortunately, no information about the bedrock is available near the calving front since this is much too crevassed to make ice radar measurements. Yet, our modelling results (see Figs. 4 and 6) show that a bump on the bed of almost half of the glacier thickness (about 125 meters) from the moraine position to the south-east glacier side can cause the observed shear on its own (see Figs. 7 and 5). Indeed a basal bump would reduce substantially the buoyancy, the basal sliding and the surface motion. Other field evidence corroborates this theory. First, the presence of a up-to-20-meter-high bump in the center of this area of the glacier surface (see Fig. 5) probably reflects an artefact in the basal topography. Interestingly, this surface hump was already reported in Chamberlin (1897), presumably because this intriguing feature was further pronounced at the end of the 19th century due to thicker ice: "The slope just beyond the crevassed area [...] and between that area and the medial moraine inclines backward [...] and a brook flows in that direction. The notable feature of the surface is the crevassing." Finally, the border side of the glacier is characterized by a ridge, which is oriented toward the sheared area (see Fig. 5). If the ridge extents under the ice, the bedrock is expected to be shallower there than elsewhere. An alternative explanation for the



high shear zone would be a locally frozen bed preventing basal sliding near the glacier margin. However, such an asymmetry in basal temperatures would likely be the result of an asymmetric bedrock. As a consequence, an intermediate case of shallower but frozen bump over the bedrock cannot be excluded, so that the 125 meters found in modelling experiments as bump height must be understood as an upper bound.

The presence of a freshwater plume at the level of the moraine (see Fig. 2a) rises the question of its role as potential trigger of the 27 July calving event. Indeed, the plume witnesses of freshwater discharge at the foot of the calving front. Yet, plumes are characterized by turbulent flow and enhanced submarine melting along the cliff. In such condition, the calving face below sea level might have been overhanging, causing the front to lean forward and favouring the opening of crevasses (Kimura et al., 2014). However, additional simulations (not shown) accounting for local overhanging of the calving front did not allow us to

establish any clear link between the presence of a plume and the opening of the main crack. Indeed, the observed jumps in the velocity field are remarkably uniform along the crack (see Fig. 2d), while digging locally the foot of the calving front locally would render the modelled jumps of velocity strong near the plume but weaker elsewhere.

Based on the analysis of the three previous paragraphs, we can now revisit chronologically the proceeding of the mechanisms, which led to the calving event of the 27th July 2015. First, a shallow and possibly frozen bedrock near the calving front between

the south-east margin and the medial moraine (see Fig. 7) slows down the ice flow in this area. In response to high resulting strain rates (see Fig. 2d), many crevasses form at the transition between the slow and fast zones. In turn, this destabilizes the vicinity of the calving front, and a major crack appears there on July 12. In contact with the sea, tidewater enters the crack, and triggers an irreversible deepening, the cryostatic pressure being no longer able to balance the water pressure at the bottom of the crevasse. The crack deepening gradually detaches the slice of ice from the glacier trunk and causes some stress

concentration at the crack tip. In turn, this causes the crack to propagate laterally over 1 km to the north-western part about 100 meter upstream of the front before to collapse on July 27 (see Fig. 1). A remaining chunk collapses on August 9 during a second major calving event. This proceeding of calving events is very similar to the set of events, which occurred in May (see Fig. 1), and we can reasonably assume that events of this type will occur again in the future as long as the calving front remains over the presumable bump. Yet, Bowdoin Glacier has experienced a rapid retreat of ca. 2 km between 2007 and 2013

before to stabilize at the current position despite a thinning at a rate of ca. 4 m/a (Tsutaki et al., 2016). Thus, we assume that the shallow and possibly frozen bedrock between the medial moraine and the south-east margin has been playing a crucial role to stabilize the calving front. Consequently, if the glaciers keeps thinning in the future, one must expect a rapid unstable retreat of the calving front shortly after this zone will be overpassed because of a likely bedrock over-deepening.

## 5  Conclusions

Using UAV photogrammetry, we captured high resolution orthoimages of the calving front of Bowdoin glacier before and after the initiation of a large fracture, which induced a major calving event. A detailed analysis of the displacement field, and the resulting strain rates allowed us to reconstruct accurately the path taken by the crack. Combined with modelling results, we could determine that the crack was more than half-thickness deep, filled of water up to sea level, and getting irreversibly



deeper when it was captured by the UAV. Later on, the crack deepening caused stress concentration around the tip, the crack to propagate laterally, and finally to collapse. Modelling results also indicated that the crack was likely triggered in a highly crevassed area near the front caused by a shallow and possibly frozen bedrock. The asymmetry of basal conditions at the front explains the calving pattern observed in May and July-August 2015, while symmetric conditions would have rendered calving

events less predictable. Importantly, our results indicate that the calving front of Bowdoin Glacier is likely stabilized by a shallow bedrock under the south-east glacier margin. As a consequence, the glacier might pass over a basal depression if the glacier keeps thinning, so that a rapid unstable retreat of the front must be expected in the next years.

In this paper, we have used UAV photogrammetry and feature tracking by closely following the techniques described by Ryan et al. (2015). By combining this approach to ice flow modelling, we could additionally analyse in details the horizontal

and vertical propagation of large water-filled crevasses near the calving front before they collapse and generate a calving event. This approach is especially relevant for large calving events given that few of them can contribute more to the global ice ablation than small and frequent ones (Medrzycka et al., 2016). However, only two snapshots of the entire fracturing process were available in the present study. In particular, this is not enough to determine whether the crack kept propagating vertically after July 16, or if the stress concentration at the tip was sufficient to allow for lateral extension as well. By contrast, frequent

UAV-inferred orthoimages and resulting flow fields between the first crack initiation and the final collapse of this particular event would have certainly allowed us to infer accurately the whole proceeding more accurately. As a consequence, future studies focusing on monitoring individual crevasses prior iceberg calving should include repetitive UAV flights to capture all phases associated to the event.

*Author contributions.* GJ designed the study, ran the Elmer/Ice model and wrote the paper with support from JS, MF and SS. GJ and YW

assembled, tuned and flew the UAV during the field campaign 2015. YW processed the aerial images by photogrammetry. TA and DS inferred the velocity fields, and the calving front positions from UAV and satellites orthoimages. HS derived the Digital Elevation Model of the bedrock from ice radar measurements. MF and SS led and organized the field campaign on Bowdoin Glacier in July 2015. All authors contributed to the paper.

*Acknowledgements.* This research was funded by the Swiss National Science Foundation, Grant 200021-153179/1, and the Japanese Ministry

of Education, Culture, Sports, Science and Technology through the GRENE Arctic Climate Research Project and the Arctic Challenge for Sustainability (ArCS) project. We thank Thomas Wyder and Rolf Jäger (eflight) for their support to construct and fly the X8 UAV, respectively.



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
