# Peer review of "Initiation of a major calving event on Bowdoin Glacier captured by UAV photogrammetry"

_The Cryosphere, 2016_

## Referee Comment (RC1) · T Zwinger (Referee) · 28 Dec 2016

**Review of: Initiation of a major calving event on Bowdoin Glacier captured by UAV photogrammetry**

by G. Jouvet et al.

December 28, 2016

**1 General impression**

This manuscript is about a joint remote sensing (from field as well as space) and modelling activity to analyse a single calving event at Bowdoin Glacier, Greenland. The observational data on the one hand comes from UAV photogrammetry as well as satellite data which both is used in a set-up of a full-stress code that analyses the dynamics from the given observational constraints. The outline of the article is clear, the text in large parts well written. Applied methods in my opinion are described well – apart from the points raised in the next section – and the graphics generally support the text.

**2 Main point of critics**

By nature of the scarce data there is a lot of assumption on ice-flow and basal sliding parameters as well as the basal geometry involved in this study. It is to be expected that the combination of those is not unique to the problem, i.e., a different sliding law with a changed ice rheology and bedrock topography could have led to a similar result. In my opinion you have to elaborate in order not to provide the grounds for interpreting justifiable assumptions as speculations.

For the geometry: I reckon that getting bathymetry at or underneath calving fronts is close to a mission impossible, as you confirm yourself in your paper. From Sugiyama et al. (2015) I understand that basically a single longitudinal profile with some lateral measurements (none close to the front) have been measured above the glacier. That is just the reality in ice sheet modelling, but it should be presented more clearly in the paper. You perhaps could realize that by indicating the measurements on a bedrock map of your final configuration, like you do for the glacier surface in Fig. 5. The only information we get on the optimized bedrock is the cross-section in Fig 7.

In case of ice flow and sliding parameters, I am more concerned about the extraordinary high rate factor of $E = 10$ as you seem to be. Simply, because even if in the region of the borehole to which you optimize this value this does not have a large influence, I would say that at the margin and particular at high deformation areas in the vicinity of the assumed undulation it could be important. Thus it would be good if you would

elaborate what could be the cause for the need to employ such a vast increase of fluidity? I can suggest three possible reasons:

1. Change of effective viscosity either by temperature or damage: For the first, you provide a not accessible reference (Seguinot et al.) but do not elaborate what temperature profile you assume for the glacier tongue, so please, add this information in this manuscript. It, though, would need a vastly warmer temperature to achieve such a strong increase of fluidity. But what about damage of the ice? How crevassed is the ice at the borehole?

2. Anisotropy: Could the missing ability to preferentially yield to vertical shear be explained by a pronounced fabric?

3. Wrong deformation to sliding ratio for sane values of $E$, caused by a wrong combination of geometry and friction law coefficients or friction law physics. There is nothing you can do about the missing geometry, but perhaps you could think of other sliding laws?

In that context: Did you rerun the global model after you sequentially optimized the remaining parameters (bump height, crevasse-depth) for the tongue model? Did it reveal a similar ratio at the borehole if applying $E = 10$ with the altered geometry?

Further, did you check in a prognostic run how the flow evolves with a given configuration? I did not find any hint on the nature of your runs in the text, so forgive me, if this went unnoticed. From my own experience I know that it is quite easy to tune a diagnostic simulation but as soon as one switches in free surface adjustment, the system can adapt to a in parts drastically changed dynamics very quickly (see e.g., Zwinger and Moore, 2009).

**3   Issues/Requests by their occurrence**

Requests for explanation or corrections sorted by page and line numbers of the document downloadable at `http://www.the-cryosphere-discuss.net/tc-2016-246/tc-2016-246.pdf`.

**Page 2, line 10:** "However, these semi-empirical approaches assume closely-spaced crevasses and do not account for the stress concentration at the tip of cracks." This could be misinterpreted as if stress accumulation would be existent in the physics, but simply is ignored in the crevasse depth criteria. So – in my opinion – rather they do not *include* or *do not have to account for* the effect of stress concentration, because of the proximity of crevasses.

**Page 5, Figure 2:** There seem to be two distinct sets of vectors in terms of directions and lengths showing along the crack. I presume they link to the opposite faces of the crack, hence the difference between them is the crack opening rate - if I am not mistaken, this would be worth explaining either in the caption, or – as

space obviously is limited – in the text referring to Figure 2. If you even could augment this information with different colours for the arrows from opposite faces. Furthermore, panel d) shows a reduced area compared to a)-c). Could you perhaps mark it with a box in c) (like you did for the inlet in b)) such that for the reader it is easier to sort the scales.

**Page 6, line 12:** "To do so, we built a 3D mesh of the glacier front from estimated basal and surface ice topographies using ..." I, personally, do not like the term *estimated* (see comment in previous section). You should clearly distinguish between the data you know and the parts you had to fill in.

**Page 7, line 5:** "A relationship between englacial temperature and depth was obtained from borehole measurements (Seguinot et al., in prep.), and generalized over the whole modelled domain." Give a short description of the temperature distribution and a reason why this can be assumed to be valid all over the glacier. This is important in relation to my major critics about the assumption you have to take on your rheology. Wrong temperatures naturally would call for a correction of the viscosity.

**Page 7, line 11:** For Equation (3) neither gravity nor density of water are defined. Reading further, I realize that the latter is defined one paragraph below - why not introduce it here?

**Page 7, line 26:** " Parameter $\alpha$ in Eq. (5) therefore controls the degree of influence of the floatation ratio on sliding from no-influence when $\alpha = 0$ (Weertman's law) to a quadratic influence when $\alpha = 2$." On page 11, line 1, you talk of "limit cases " (which I would write as "limiting cases"), so, please, explain why you exactly set $\alpha = 2$ as the upper limit.

**Page 8, line 1:** "Since the borehole is located about 2 km upstream the calving front (see Fig. 3), we first modelled an enlarged domain, and used satellite-inferred velocity field (see Fig. 3) instead of the UAV-inferred ones since it covers a larger area. Once the three ice flow parameters were tuned, we restricted our domain to the glacier front, and optimized (to improve the consistency between modelled and measured velocity fields) according to two geometrical parameters $h_{\text{lift}}$, and $d_{\text{frac}}$ of the glacier front (see Fig. 4, middle and bottom panels)" . If you cut through the glacier (I guess Fig. 6 is representative to your reduced size model), in my view two questions should be addressed in the text:

1. What boundary conditions for inflow velocities and stresses (which are interlinked) you do assume for the in size reduced model?

2. As you seem to optimize with respect to two different sets of surface velocities: Are the satellite velocities referenced to the UAV velocities where they overlap?

**Page 12, line 19:** "Because the crack was connected to the sea (see Fig. 2a), the water level within the crevasse most likely coincided with the sea level, or was slightly above." How can a to the ocean connected crevasse maintain a water level above the sea level?

**Page 12, line 20:** "On the other side, the tensile *force* must have been in the order of magnitude of critical stress thresholds, i.e. around 0.1-0.3 MPa (Vaughan, 1993). Yet, the crack is expected to deepen further if the pressure of water plus the tensile *force* exceed the pressure of ice (Van der Veen, 1998)." You are writing of "tensile force", but I guess you mean "tensile stress", as you compare to MPa units.

**Page 13, Figure 7.:** Could you consider using another colour for highlighting the crack position, as it is not really distinguishable from the middle moraine? In connection to the lower figure I have two questions:

1. Why is there a solid line at the place of the apparently optimized bedrock at the bump?
2. How is the apparently varying crack depth corresponding to your initial assumption of a constant crack depth?

A further suggestion: Could you clearer distinct the dashed patterns for crack depth and optimized bedrock? Perhaps use a dashed-dotted line for one of them.

**4 Typos, etc.**

Typos and suggestions about typesetting – like before – sorted by their occurrence:

**Page 2, line 11:** "(Astrom et al., 2013)". $\rightarrow$ (Åström et al., 2013); Please, correct this also in the citation (NB: Swedish Å typeset in LaTeX as \{AA}).

**Page 2, line 28:** "...in term of area". $\rightarrow$ ...in terms of area.

**Page 6, Figure 2:** "...with a magnitude above 5 $a^{-1}$ are drawn on d)". The unit should be written in normal text, not as formula.

**Page 7, line 3:** "..., where $\epsilon$ and $\sigma$ denote the strain and deviatoric stress tensor, $\sigma_{II}$ denotes the second invariant of $\sigma$, ..." Please, check with the guidelines of the journal, whether – like often practiced – not indexed tensors have to be written in bold symbols, such as $\boldsymbol{\sigma}$ and $\boldsymbol{\epsilon}$.

**Page 7, line 15:** "At the glacier bed, one applies Budd's friction law (Budd et al., 1984), ..." Admittedly, it is a matter of taste how to formulate it, but I would replace "one applies" with "we apply". In connection to my general statement, it would be good to justify why exactly Budd's law and no alternative to it would be the right choice.

**Page 2, line 11:** "...to mimic with a presumable bump ..." Perhaps better: "...to mimic an assumed bump ..."

**Page 8, line 10:** "...(assuming it is filled of water up to sea level, see Section 4)..." Would perhaps "...(assuming it is filled *with* water up to sea level, see Section 4)..." be correct?

**Page 8, line 26:** "..., which monitors the calving front at about 2 km distance." Perhaps better: "..., which monitors the calving front *from* about 2 km distance."

**Page 12, line 11:** "..., and is only hanging to the other side." Suggestion: "..., and remains only attached to the other side."

**Page 12, line 25:** "..., and caused the crack to extend further laterally before to collapse (see Fig. 1)." Did you mean: "..., and caused the crack to extend further laterally before *the* collapse (see Fig. 1)." ?

**Page 13, line 10:** " First, the presence of a up-to-20-meter-high bump in the center of this area of the glacier surface (see Fig. 5) probably reflects an artefact in the basal topography.." To me an "artefact" is a non-existing misleading deviation in observational data. Would "feature" be a better choice of word?

**Page 17, line 23:** "Seguinot, J., Funk, M., Jouvet, G., Ryser, C., Bauder, A., Leinss, S., D., S., and Sugiyama, S.: Tidewater glacier dynamics dominated by sliding at Bowdoin Glacier, Northwest Greenland, in prep, in prep." drop the second "in prep". My personal opinion is that at most submitted papers should be cited – perhaps we can get some suggestions from the editor/journal side on this, or the cited paper in the next revision of this one is able to be linked to at least a journal.

**References**

Sugiyama, S., Sakakibara, D., Tsutaki, S., Maruyama, M., and Sawagaki, T.: Glacier dynamics near the calving front of Bowdoin Glacier, northwestern Greenland, Journal of Glaciology, 61, 223–232, doi:10.31892015JoG14J127, 2015.

Seguinot, J., Funk, M., Jouvet, G., Ryser, C., Bauder, A., Leinss, S., D., S., and Sugiyama, S.: Tidewater glacier dynamics dominated by sliding at Bowdoin Glacier, Northwest Greenland, in prep. (not accessible to the reviewer)

Zwinger T. and J.C. Moore, 2009. Diagnostic and prognostic simulations with a full Stokes model accounting for superimposed ice of Midtre Lovénbreen, Svalbard, The Cryosphere, 3, 217-229, doi:10.5194/tc-3-217-2009

---

## Referee Comment (RC2) · Anonymous Referee #2 · 25 Jan 2017

Summary In this paper the authors combine remote sensing with satellite data and aerial photogrammetry as well as modelling using ELMER/ICE to analyse a calving event on the Bowdoin Glacier in north eastern Greenland. Their observations from before and after the event allow them to make inferences about both the way the crack propagated and the likely future retreat of the glacier. Furthermore, the authors consider this a useful case study for similar glaciological investigations in the future. Comments This is a very nice example of a study that uses many different techniques, some of them fairly novel, such as the use of the UAV to derive very detailed strain rates, in order to clarify an important glaciological process. In this case, the process of crack initiation and calving are studied on a glacier that is representative of many Greenlandic outlet glaciers. The authors have clearly thought hard about how to introduce and structure this paper such that the disparate techniques are well explained and build a

consistent picture of the whole. The paper is very well written and easy to follow, there are a few typos + English language errors which I list below. I have no hesitation in recommending this paper for publication however there are a few minor corrections I would suggest. Improvements: 1. The retreat of the Bowdoin Glacier is undoubtedly interesting, particularly given different timings of retreat from local and regional glaciers so I think it would be helpful to point out that while the general retreat around Greenland is triggered by warming temperatures, local conditions determine the timing and pace of retreat. There is now quite a lot of evidence from temperature data in Greenland and fairly long records are available for Thule/Pituffik but I think reference to one of the many studies that have been done Kjeldsen et al for example would be fine. 2. On a similar point, you mention in the discussion that the crack propagated down after the 16th July – possibly in response to a change in water level. Have you looked to see if this is from meltwater? July is the middle of the ablation season and there should be a lot of meltwater going in. It would be neat to see if the calving activity matches up well with the melt activity 3. On page 12, line 26 you state Nye 1957 had an analytical solution for water in crevasses. I'm pretty sure Nye's 1957 paper did not include water in the crevasses in the analytical solution but I could be wrong and can't find my copy of it right now so do check this. It is something we developed in the Benn et al 2007 paper to an extent and I also did quite a lot of modelling in my thesis of this but it's not been published and as you say here the stress concentration at the crack tip effect is a problem. However Hans Weertman's work (1974,1977 and chapter 10 in his book) are relevant here and might prove interesting to further study. I do not necessarily think you should go into this here though but do check the reference! Typos: p.2 line 28 "terms" p.3. line 10 "upstream of its" p.7 line 18 "flotation" not floatatio p.8 line 6 "to mimic a bump presumed responsible.." line20 "north-west tens of days later" p.12 line 10 "issue, and confirms that the fracture extends to the front" p.12 line 26: "we cannot corroborate" line 27 "since Nye's formula" line 29 "upstream of the junction" p.13 line 16 "extends" p.14 line 6 "the plume suggests that there is" or "the plume shows that there is" line 11 doesn't make sense - while digging locally the foot of the calving front
locally ?

---

## Author Comment (AC1) · 24 Feb 2017

Summary: In this paper the authors combine remote sensing with satellite data and aerial photogrammetry as well as modelling using ELMER/ICE to analyse a calving event on the Bowdoin Glacier in north eastern Greenland. Their observations from before and after the event allow them to make inferences about both the way the crack propagated and the likely future retreat of the glacier. Furthermore, the authors consider this a useful case study for similar glaciological investigations in the future. Comments This is a very nice example of a study that uses many different techniques, some of them fairly novel, such as the use of the UAV to derive very detailed strain rates, in order to clarify an important glaciological process. In this case, the process of crack initiation and calving are studied on a glacier that is representative of many Greenlandic outlet glaciers. The authors have clearly thought hard about how to intro-

duce and structure this paper such that the disparate techniques are well explained and build a consistent picture of the whole. The paper is very well written and easy to follow, there are a few typos + English language errors which I list below. I have no hesitation in recommending this paper for publication however there are a few minor corrections I would suggest.

Improvements: 1. The retreat of the Bowdoin Glacier is undoubtedly interesting, particularly given different timings of retreat from local and regional glaciers so I think it would be helpful to point out that while the general retreat around Greenland is triggered by warming temperatures, local conditions determine the timing and pace of retreat. There is now quite a lot of evidence from temperature data in Greenland and fairly long records are available for Thule/Pituffik but I think reference to one of the many studies that have been done Kjeldsen et al for example would be fine.

ANSWER: We added after the first sentence of the introduction the two following sentences: "While the general retreat is triggered by warming temperatures, local conditions determine the timing and pace of retreat (Kjeldsen and al., 2015). For instance, tidewater glaciers of the north-western Greenland started rapid retreat in recent years only, i.e. about twenty years after glaciers of the southern."

2. On a similar point, you mention in the discussion that the crack propagated down after the 16th July – possibly in response to a change in water level. Have you looked to see if this is from meltwater? July is the middle of the ablation season and there should be a lot of meltwater going in. It would be neat to see if the calving activity matches up well with the melt activity.

ANSWER: It is true that the two weeks proceeding the 16th July were characterized by relatively warm temperatures, and cloudless sky. Therefore, the period must has been accompanied with intense meltwater production. We added the following sentence in the first paragraph of the discussion section: "Such an increase could have been triggered by intense meltwater production during the first half of July, consistently to warm temperatures, and cloudless sky observed on the field."

3. On page 12, line 26 you state Nye 1957 had an analytical solution for water in crevasses. I'm pretty sure Nye's 1957 paper did not include water in the crevasses in the analytical solution but I could be wrong and can't find my copy of it right now so do check this. It is something we developed in the Benn et al 2007 paper to an extent and I also did quite a lot of modelling in my thesis of this but it's not been published and as you say here the stress concentration at the crack tip effect is a problem. However Hans Weertman's work (1974,1977 and chapter 10 in his book) are relevant here and might prove interesting to further study. I do not necessarily think you should go into this here though but do check the reference!

ANSWER: Thank you for pointing this. We changed the reference to (Benn et al, 2007).

**Typos:**
p.2 line 28 "terms"
p.3. line 10 "upstream of its"
p.7 line 18 "flotation" not floatation
p.8 line 6 "to mimic a bump presumed responsible.."
line 20 "north-west tens of days later"
p.12 line 10 "issue, and confirms that the fracture extends to the front"
p.12 line 26: "we cannot corroborate"
line 27 "since Nye's formula"

line 29 "upstream of the junction"
p.13 line 16 "extends"
p.14 line 6 "the plume suggests that there is" or "the plume shows that there is"
line 11 doesn't make sense - while digging locally the foot of the calving front locally ?

ANSWER: All typos were corrected in the revised manuscript.

---

## Author Comment (AC2) · 24 Feb 2017

NOTE: The revised version of the manuscript is attached in supplement. All the corrections made appear in red.

This manuscript is about a joint remote sensing (from field as well as space) and modelling activity to analyse a single calving event at Bowdoin Glacier, Greenland. The observational data on the one hand comes from UAV photogrammetry as well as satellite data which both is used in a set-up of a full-stress code that analyses the dynamics from the given observational constraints. The outline of the article is clear, the text in large parts well written. Applied methods in my opinion are described well apart from the points raised in the next section and the graphics generally support the text.

**2 Main point of critics**

By nature of the scarce data there is a lot of assumption on ice-flow and basal sliding parameters as well as the basal geometry involved in this study. It is to be expected that the combination of those is not unique to the problem, i.e., a different sliding law with a changed ice rheology and bedrock topography could have led to a similar result. In my opinion you have to elaborate in order not to provide the grounds for interpreting justifiable assumptions as speculations.

For the geometry: I reckon that getting bathymetry at or underneath calving fronts is close to a mission impossible, as you confirm yourself in your paper. From Sugiyama et al. (2015) I understand that basically a single longitudinal profile with some lateral measurements (none close to the front) have been measured above the glacier. That is just the reality in ice sheet modelling, but it should be presented more clearly in the paper. You perhaps could realize that by indicating the measurements on a bedrock map of your final configuration, like you do for the glacier surface in Fig. 5. The only information we get on the optimized bedrock is the cross-section in Fig 7.

ANSWER: We included two additional maps to Fig. 5, with the level lines of the initial and optimized bedrock.

In case of ice flow and sliding parameters, I am more concerned about the extraordinary high rate factor of E = 10 as you seem to be. Simply, because even if in the region of the borehole to which you optimize this value this does not have a large influence, I would say that at the margin and particular at high deformation areas in the vicinity of the assumed undulation it could be important. Thus it would be good if you would elaborate what could be the cause for the need to employ such a vast increase of fluidity?

ANSWER: Thank you for pointing this issue. After some checks, we actually found an error in the input files for Elmer/Ice, and all the results we have shown in the first manuscript assumed a constant ice temperature of $-10°$. This explains why we had to set a very high enhancement factor of $E = 10$ to compensate. This is now fixed, all models have been run again, the new optimal enhancement factor is $E = 4$ to produce $10\%$ of vertical deformation at the borehole location, consistently to measurements. Results about the optimal crack depth have been updated. Still, we tested other enhancement factors $E = 1, 2, 8$. We found that their impacts on optimal bed uplift and crack depth variables was rather limited, and we included this new finding in the revised manuscript.

I can suggest three possible reasons:

- 1. Change of effective viscosity either by temperature or damage: For the first, you provide a not accessible reference (Seguinot et al.) but do not elaborate what temperature profile you assume for the glacier tongue, so please, add this information in this manuscript. It, though, would need a vastly warmer temperature to achieve such a strong increase of fluidity. But what about damage of the ice? How crevassed is the ice at the borehole?

  ANSWER: To answer your question, we added one section called "Borehole temperature and deformation data", in which we elaborate the temperature profile available from borehole measurements, and removed the reference to (Seguinot et al., in prep.). Damage ice is tested via additional runs with different enhancement factors ($E = 1, 2, 8$).

- 2. Anisotropy: Could the missing ability to preferentially yield to vertical shear be explained by a pronounced fabric?

ANSWER: Possible anisotropic ice is also tested through the additional runs with different enhancement factors.

- 3. Wrong deformation to sliding ratio for sane values of E, caused by a wrong combination of geometry and friction law coefficients or friction law physics. There is nothing you can do about the missing geometry, but perhaps you could think of other sliding laws?

ANSWER: The sliding law we opted for is the most elaborate law with the input data we have. Downgrading it to a Weertmann-like law (i.e. removing the dependency on the effective pressure) or upgrading it to a Coulomb-like friction law would likely not impact our results since the sliding coefficient is in any case tuned to measured velocities.

In that context: Did you rerun the global model after you sequentially optimized the remaining parameters (bump height, crevasse-depth) for the tongue model? Did it reveal a similar ratio at the borehole if applying E = 10 with the altered geometry?

ANSWER: After some checks, we found that the $90\%$ sliding/shearing ratio at the borehole is not affected by local changes (bump height, crevasse-depth) near the front: the borehole is too far (about 2 km) to be influenced.

Further, did you check in a prognostic run how the flow evolves with a given configuration? I did not find any hint on the nature of your runs in the text, so forgive me, if this went unnoticed. From my own experience I know that it is quite easy to tune a diagnostic simulation but as soon as one switches in free surface adjustment, the system can adapt to a in parts drastically changed dynamics very quickly (see e.g., Zwinger and Moore, 2009).

ANSWER: No, we haven't tried any prognostic runs. Yet, our free surface is nearly flat. Therefore, we do not expect much adjustments after running Elmer/Ice forward-in-time.

**3 Issues/Requests by their occurrence**

Page 2, line 10: However, these semi-empirical approaches assume closely-spaced crevasses and do not account for the stress concentration at the tip of cracks." This could be misinterpreted as if stress accumulation would be existent in the physics, but simply is ignored in the crevasse depth criteria. So in my opinion rather they do not include or do not have to account for the effect of stress concentration, because of the proximity of crevasses.

ANSWER: We rephrased the sentence as follows: "However, these semi-empirical approaches assume closely-spaced crevasses and do not include the effect of the stress concentration at the tip of cracks."

Page 5, Figure 2: There seem to be two distinct sets of vectors in terms of directions and lengths showing along the crack. I presume they link to the opposite faces of the crack, hence the difference between them is the crack opening rate - if I am not mistaken, this would be worth explaining either in the caption, or as space obviously is limited in the text referring to Figure 2. If you even could augment this information with different colours for the arrows from opposite faces. Furthermore, panel d) shows a reduced area compared to a)-c). Could you perhaps mark it with a box in c) (like you did for the inlet in b)) such that for the reader it is easier to sort the scales.

ANSWER: This is correct. We coloured differently the arrows of each face of the crack, included this feature to the caption, and added a box on panel c) to ease the reading of panel d).

Page 6, line 12: To do so, we built a 3D mesh of the glacier front from estimated basal
and surface ice topographies using ..." I, personally, do not like the term estimated
(see comment in previous section). You should clearly distinguish between the data
you know and the parts you had to fill in.

ANSWER: The word "estimated" was removed, the "basal topography" is defined
and clarified in the next sentence.

Page 7, line 5: A relationship between englacial temperature and depth was obtained
from borehole measurements (Seguinot et al., in prep.), and generalized over the whole
modelled domain." Give a short description of the temperature distribution and a reason
why this can be assumed to be valid all over the glacier. This is important in relation
to my major critics about the assumption you have to take on your rheology. Wrong
temperatures naturally would call for a correction of the viscosity.

ANSWER: As said previously, we added one section called "Borehole tempera-
ture and deformation data", in which we elaborate the temperature profile available
from borehole measurements. Our additional runs based on enhancement factors
$E = 1, 2, 8$ allow us to explore several ice fluidities, and therefore to account for
uncertainties in the englacial temperature data.

Page 7, line 11: For Equation (3) neither gravity nor density of water are defined. Read-
ing further, I realize that the latter is defined one paragraph below - why not introduce
it here?

ANSWER: Gravity and water density are now defined earlier in the text, as it should
be.

Page 7, line 26: "Parameter $\alpha$ in Eq. (5) therefore controls the degree of influence
of the floatation ratio on sliding from no-influence when $\alpha = 0$ (Weertman's law) to a

quadratic influence when $\alpha = 2$." On page 11, line 1, you talk of limit cases (which I would write as limiting cases"), so, please, explain why you exactly set $\alpha = 2$ as the upper limit.

ANSWER: We agree that the limit $\alpha = 2$ is somewhat arbitrary. Yet, values between 0 to 2 for effective pressure-dependent sliding laws are most often reported in the literature, e.g. $2$ in () or $3.5/3$ in (). We clarified our choice of limit values for $\alpha$ in the revised manuscript.

Page 8, line 1: Since the borehole is located about 2 km upstream the calving front (see Fig. 3), we first modelled an enlarged domain, and used satellite-inferred velocity field (see Fig. 3) instead of the UAV-inferred ones since it covers a larger area. Once the three ice flow parameters were tuned, we restricted our domain to the glacier front, and optimized (to improve the consistency between modelled and measured velocity fields) according to two geometrical parameters $h_{lift}$, and $d_{frac}$ of the glacier front (see Fig. 4, middle and bottom panels)". If you cut through the glacier (I guess Fig. 6 is representative to your reduced size model), in my view two questions should be addressed in the text:

- 1. What boundary conditions for in flow velocities and stresses (which are interlinked) you do assume for the in size reduced model?

  ANSWER: The boundary conditions in the large and reduced size models are strictly the same. The only difference is the input velocities data, which are supplied from satellites images for the large model and from UAV images for the reduced one. For the sake of clarification, we drew the limits of the two domains on Fig. 3, and wrote specifically that the same boundary conditions apply to both modelled domains.

- 2. As you seem to optimize with respect to two different sets of surface velocities: Are the satellite velocities referenced to the UAV velocities where they overlap?

  ANSWER: No, the two velocity fields were computed independently from satellite and UAV images. Yet, they were found to be consistent over the overlapping part, as I wrote in the second sentence of section 3.3.

Page 12, line 19: Because the crack was connected to the sea (see Fig. 2a), the water level within the crevasse most likely coincided with the sea level, or was slightly above." How can a to the ocean connected crevasse maintain a water level above the sea level?

  ANSWER: This statement was based on the hypothesis of a supra-glacial torrent discharging into the crack (as observed during our fieldwork), which could have maintained the water level in the crack slightly above sea level. Yet, we revised our manuscript by removing "or was slightly above", since this hypothesis is insufficiently founded.

Page 12, line 20: On the other side, the tensile force must have been in the order of magnitude of critical stress thresholds, i.e. around 0.1-0.3 MPa (Vaughan, 1993). Yet, the crack is expected to deepen further if the pressure of water plus the tensile force exceed the pressure of ice (Van der Veen, 1998)." You are writing of tensile force", but I guess you mean tensile stress", as you compare to MPa units.

  ANSWER: You are right. This is corrected.

Page 13, Figure 7.: Could you consider using another colour for highlighting the crack position, as it is not really distinguishable from the middle moraine? In connection to the lower figure I have two questions:

- 1. Why is there a solid line at the place of the apparently optimized bedrock at the bump?

- 2. How is the apparently varying crack depth corresponding to your initial assumption of a constant crack depth? A further suggestion: Could you clearer distinct the dashed patterns for crack depth and optimized bedrock? Perhaps use a dashed-dotted line for one of them.

ANSWER: We highlighted the crack with a dashed line on the top panel, fixed the solid line issue, added varying crack depths to the bottom panel, and better distincted the dashed patterns for crack depth and optimized bedrock.

**4 Typos, etc.**

ANSWER: All typos were corrected, and the reference (Seguinot, in prep.) was removed.

[Figure]

**Supplement:**

[revised manuscript text omitted]